# Viral Metagenomics as a Tool to Track Sources of Fecal Contamination: A One Health Approach

**DOI:** 10.3390/v15010236

**Published:** 2023-01-14

**Authors:** Tasha M. Santiago-Rodriguez, Emily B. Hollister

**Affiliations:** Diversigen, New Brighton, MN 55112, USA

**Keywords:** fecal contamination, One Health, source tracking, viral detection, viral metagenomics

## Abstract

The One Health framework recognizes that human, animal, and environmental health are linked and highly interdependent. Fecal contamination of water, soil, foodstuff, and air may impact many aspects of One Health, and culture, PCR-based, and sequencing methods are utilized in the detection of fecal contamination to determine source, load, and risk to inform targeted mitigation strategies. Viruses, particularly, have been considered as fecal contamination indicators given the narrow host range many exhibit and their association with other biological contaminants. Culture- and molecular-based methods are considered the gold-standards for virus detection and for determining specific sources of fecal contamination via viral indicators. However, viral metagenomics is also being considered as a tool for tracking sources of fecal contamination. In the present review, studies tracking potential sources of fecal contamination in freshwaters, marine waters, foodstuff, soil, and air using viral metagenomics are discussed to highlight the potential of viral metagenomics for optimizing fecal source tracking. Limitations of the use of viral metagenomics to track fecal contamination sources, including sample processing, nucleic acid recovery, sequencing depth, and bioinformatics are also discussed. Finally, the present review discusses the potential of viral metagenomics as part of the toolbox of methods in a One Health approach.

## 1. Introduction

Water sources used for consumption, recreation, and irrigation are essential to human, animal, plant, and environmental health, but can become contaminated with fecal material originating from anthopogenic sources (e.g., wastewater and animal farms) [1]. Ingestion of and contact with fecally-contaminated water, both directly and indirectly, affects millions of people every year worldwide, resulting in gastrointestinal and respiratory illnesses, as well as eye and skin infections caused by enteric pathogens [2]. Both human and animal sources of fecal contamination represent major concerns to public health as enteric pathogenic bacteria, protozoans, and enteric viruses can be transmitted through water sources (waterborne), and cause diseases such as diarrhea, cholera, typhoid, and polio, resulting in thousands of deaths each year [3]. Fecal contamination can also directly and indirectly affect soils and foodstuff via runoff, or in cases where the water used for irrigation is contaminated with fecal material as a result of leakage or other unintentional discharge from wastewater systems, septic tanks, or pipes. This may result in foodborne diseases, food recall, decaying environmental health, and economic loss [4]. Similarly, fecally-contaminated water sources may affect livestock health and well-being, as a result of drinking microbially-polluted water and grazing on contaminated grasses [5]. In addition, aging wastewater treatment infrastructure, increased prevalence of treatment- and removal-resistant pathogens, increased need for recreational and drinking water sources, and climate change may contribute to an increase in waterborne and foodborne disease transmission [6]. Thus, it is apparent that fecal contamination affect many aspects of One Health, a concept that recognizes that human, animal, and environmental health are all interconnected [7].

Applying a One Health approach enables the consideration of both pathogenic and non-pathogenic microbes originating from the gastrointestinal tract of warm-blooded animals, and their transfer between humans, animals, and the environment. Among the potential enteric pathogens that can be introduced into water sources used for consumption, recreation, and irrigation, viruses are of great concern for several reasons. For instance, enteric pathogenic viruses are associated with hepatitis, diarrhea and gastroenteritis, as well as respiratory, eye, and skin infections [8]. In addition to humans, enteric virus infections can have detrimental effects on animal health by causing pregnancy loss, disease, and mortality [9]. Enteric viruses can also be easily transmitted through food, water, and fomites, and can cause disease with low infectious doses [10]. For instance, it has been estimated that the presence of one rotavirus plaque forming unit (PFU) has nearly 30% probability of leading to infection [11]. Moreover, enteric pathogenic viruses can be resistant to conventional treatments such as chlorination and filtration, as well as to a wide pH range (pH 3 to 10) compared to bacteria [8,12]. Finally, the survival and inactivation rates of enteric pathogenic viruses are also usually of concern. For instance, the time required for 90% of an initial inoculum of Poliovirus 1 to inactivate in seawater depends on both season and incubation temperature, and can range from 0.8 to 10 days [13]. Freshwater studies using polioviruses, echoviruses, and coxsackieviruses have shown inactivation rates of less than 1 log_10_ per day under specific experimental conditions in river water, unpolluted river water, impounded water, and ground water, suggesting that these viruses could survive for prolonged periods in fresh water environments [14,15]. This is particularly important when it comes to wastewater treatment effluent-receiving environments, as viruses can often be ignored as quality indicators in the context of regulatory standards for wastewater treatment plants. In soil samples, some of the longest reported survival times have been up to 11 and 96 days during summer and winter, respectively [14]. Notably, virus survival and inactivation rates are dependent upon the type of virus, as well as environmental conditions including temperature, pH, season, type of matrix (both organic and inorganic) [16], and autochthonous microbiota [14].

Identifying the origin and extent of fecal contamination in water sources, soil, foodstuff, and air is necessary to minimize the potential risks that enteric pathogens (as well as other pathogenic species and antibiotic-resistance genes (ARGs)) may pose to human, animal, and environmental health. Since many enteric viruses, both pathogenic and non-pathogenic (i.e., bacteriophages) can be broadly host-specific, they are considered suitable indicators to track and monitor specific sources of fecal contamination [17]. For instance, host-specific enteroviruses, including bovine and porcine enteroviruses, have been detected in water sources suspected to be contaminated with these respective sources of fecal contamination [18,19]. Similarly, human-specific pathogenic viruses, such as human adenoviruses, have also been identified in water sources including fresh, coastal, and drinking water, and have been associated with disease outbreaks [20]. Bacteriophages have also been explored as indicators of specific sources of fecal contamination. For instance, coliphages, both somatic and F+ RNA coliphages [21,22], *Bacteroides fragilis* phages [23], *Enterococus faecalis* phages (enterophages) [24], and more recently, crAssphages, which are known to infect *Bacteroides* [25], have been shown to have several of the characteristics of an ideal indicator of specific sources of fecal contamination. Several of these characteristics include limited host range due mostly to strain specificity, and similar survival and inactivation rates compared to pathogenic enteric viruses. 

## 2. Methods for Virus Detection to Track Sources of Fecal Contamination

To identify, and ultimately remediate sources of fecal contamination and avoid disease, enteric pathogenic viruses may be detected using culture- and/or molecular-based methods. More recently, and perhaps less frequently, viral metagenomics has also been explored as a tool to characterize the resident virome of a specific sample, as well as those viruses associated with specific hosts that may have been introduced as a result of a fecal contamination event. Indeed, many of the methods employed in environmental virology have been adapted from the clinical setting, with the caveat that the methods often need to be modified in order to detect the low viral loads commonly present in certain environmental samples. In the following section, culture, and molecular methods, as well as viral metagenomics for virus detection in environmental samples are summarized and discussed (Figure 1). 

### 2.1. Culture Assays

Culture methods have long been the gold standard for the identification and characterization of viruses from various sources including, but not limited to, environmental, animal, and clinical samples. Culture assays include the use of cells and tissue to identify, monitor, and propagate human and animal viruses (Figure 1). Cells lines such as Buffalo green monkey kidney (BGMK), Caco-2, HeLa, and human embryonic kidney (HEK) cells have been used to identify enteric pathogenic viruses by observing cytopathic effects (CPE) [26]. Similarly, liquid media and agar, and specific susceptible bacterial strains are needed for the identification and propagation of bacteriophages. Lysis zones on agar plates, or no growth in liquid media, may then be observed to identify phage activity using culture methods. Media and conditions needed for phage propagation will be dependent on optimal replication temperature, pH, and the physiology of the bacterial host [27]. For instance, Tryptic Soy Broth (TSB) and Agar (TSA), with CaCl_2_ (for better phage adsorption), and NaN_3_ (to minimize background growth of other bacteria) have been used for the detection of *Enterococcus faecalis* phages in recreational waters [24]. Notably, culture methods are frequently supplemented with molecular methods given that culture methods can be time-consuming and may require extensive periods of time to obtain results or several passages to confirm results, and not all enteric pathogen viruses are cultivable or show CPE [28].

### 2.2. Molecular Methods

Advancements in genome sequencing have enabled the development of molecular methods for the detection of viruses. Molecular-based methods have significantly decreased the time required for virus detection, while improving sensitivity by targeting specific, unique, and conserved genomic viral regions. Indeed, several water quality guideline standards are implementing molecular methods due to their faster turn-around time [29,30], which allows for faster decision making and can, in turn, potentially reduce risks associated with exposure to fecally-contaminated water sources and foodstuff. 

Some of the most common molecular-based assays for virus detection in environmental samples include PCR (qualitative; targets one virus at a time), real-time quantitative PCR or qPCR (quantitative; targets one virus at a time), and reverse-transcription quantitative PCR or RT-qPCR (quantitative; targets RNA), which have been extensively reviewed elsewhere (Figure 1) [28]. Multiplex PCR and qPCR have also been tested to simultaneously detect different viruses [28]. However, as with culture methods, molecular-based techniques also may exhibit several drawbacks. For instance, it is relatively difficult to determine viral viability by using molecular-based methods [31]. Moreover, given that many different human enteric viral pathogens can be introduced into water sources, it would be impractical to detect them all simultaneously using targeted molecular methods. This also indicates that a degree of a priori knowledge is needed to apply the most suitable primers for the detection of specific viruses using PCR-based methods. For this reason, an increasing number of studies have explored viral metagenomic sequencing to identify the approximate viral composition of any sample type, identify enteric viruses, and thus, identify potential sources of fecal contamination and develop source-specific indicators. 

### 2.3. Viral Metagenomics and Single Virus Genomics

Viral metagenomics refers to the sequencing and analysis of DNA and/or RNA originating from viruses, also known as the virome. The first study documenting the use of viral metagenomics was published in 2002, where the technique was applied to characterize marine viromes [32], and numerous discoveries have been made using viral metagenomics over the last 20 years [33]. In addition, viral metagenomics has enabled the development of various applications that range from diagnostics to microbial source tracking, monitoring, and surveillance [33]. Generally, viral metagenomic sequencing is similar to microbiome sequencing, which usually targets bacteria and may include, but is not limited to, sample collection, sample processing, sequencing, and bioinformatics (Figure 1). Viral metagenomics, however, may require the isolation of viruses and viral-like particles (VLPs) to increase the viral signal, by using methods such as, but not limited to, centrifugation to remove debris, as well as filtration to remove bacterial cells. These steps are then followed by extraction of viral nucleic acids, sequencing, and bioinformatic analyses, the latter which may include the use of tools specifically designed for virome datasets [34]. Bioinformatic analyses may be preceded by sequence assembly, which may pose several advantages compared to the analysis of unassembled reads, including shorter computational times during annotation [35].

An increasing number of studies have evaluated the prevalence of specific viruses in various environments, including fecal material, using viral metagenomics. One classic example is the evaluation of crAssphage as a potential marker of human fecal contamination using viral metagenomics. crAssphage was originally discovered from human gut metagenomes [36], and initial results showed that crAssphage is prevalent across human gut samples, but more so in industrialized and Westernized human societies [37]. While qPCR primers have been developed and evaluated with great success for the detection of crAssphage in water sources and sewage [38,39,40,41], several studies have evaluated the prevalence of this bacteriophage using mapping-based approaches, where metagenomes are mapped against reference crAssphage genomes [42,43]. Similarly, other viruses have been detected using mapping-based approaches. For instance, a study searched for bacteriophage ΦB124-14 sequences in metagenomes originating from human, animal, and environmental sources [44]. ΦB124-14 is a phage that infects human-associated *Bacteroides fragilis* strains known to have source tracking potential [44]. Sequences from bacteriophage ΦB124-14 were found in assembled metagenomes originating from human-associated environments [44]. 

While viral metagenomics enables the characterization of viruses that are unculturable, one drawback of this technique for tracking sources of fecal contamination is that results may be limited to viruses present in databases, many of which are model bacteriophages and pathogenic viruses [45]. For this reason, techniques such as single-virus genomics (SVGs) have emerged to facilitate the characterization of unculturable and uncharacterized viruses that are not present in databases. Flow cytometry is utilized in SVGs to separate or sort viruses, which are fixed using a fixative agent such as glutaraldehyde, and fluorescently stained [46]. SVGs has the potential to overcome several of the intrinsic biases associated with viral metagenomics including assembly and annotation, particularly of closely related viral strains, since the technique can capture the genetic diversity of uncultured viruses [46]. To date, no studies have employed SVGs as a tool to track sources of fecal contamination. However, previous studies have applied this technique to characterize viruses from oceans and discriminate between closely related viruses [46], suggesting its potential as a technique for tracking sources of fecal contamination. 

## 3. Evaluation of Viral Metagenomics for Tracking Sources of Fecal Contamination

Fecal contamination of water sources, foodstuff, soil, and air, and the introduction of viruses into these environments, may impact many aspects of One Health, particularly through the potential detrimental effects on human and animal health (Figure 2). Viral metagenomics has been explored as a tool to track sources of fecal contamination and determine the presence of enteric viruses in various sample types. Examples of how these studies have been used to track sources of fecal contamination in specific environments, particularly freshwaters, marine waters, wastewater, fecal material, foodstuff, soil, and air are described in the sections below. 

### 3.1. Freshwaters and Marine Waters

Freshwater and marine water are essential for drinking, recreation, and irrigation purposes and include, but are not limited to, beaches, ponds, lakes, wetlands, and rivers. Viral metagenomics has also been applied to track sources of fecal contamination in freshwaters and marine waters (Figure 2). The following section describes applications and specific examples of the use of viral metagenomics as a tool to track sources of fecal contamination in freshwaters and marine waters. 

Rivers are essential resources in urban and rural areas as they provide water for livelihood, irrigation, landscape, and recreation [47]. Unfortunately, rivers, like many freshwater sources, receive significant waste that may contain pathogenic enteric viruses; thus, identifying potential sources of fecal contamination is essential. A recent study evaluated a toolbox of bioinformatic methods used in determining fecal contamination and the presence of ARGs in a peri-urban river [43]. In this study, sediment and water samples were collected from the peri-urban river, as were sewage influent, effluent, and chicken and pig manure. Shotgun metagenomic sequencing targeting the global microbiome (i.e., bacteria, viruses, and fungi) was applied to each of these sample types. BLASTp against the Deep-ARG database was applied to determine the prevalence of ARGs in the collected samples, and the presence of crAssphage was determined by mapping sequence reads to a collection of virus reference genomes [48]. Finally, SourceTracker, a machine-learning tool commonly used to quantify the potential contributions of specific sources to a community profile was also evaluated [49]. Over 800 ARGs belonging to >25 ARG types were detected across the samples, with ARG prevalence corresponding to the extent of the human fecal contamination, as determined by the SourceTracker and crAssphage results. Although other viral indicators of fecal contamination, including *Bacteroides* phage ΦB124-14, human adenovirus F (HAdV), human polyomavirus BK (HPyV), and pepper mild mottle virus (PMMoV) were not detected among the river sediments, studies such as this provide new insights into the application of viral and/or ARG profiling to understand fecal contamination in peri-urban ecosystems, and other freshwater systems, and the relationship between these two indicators of fecal contamination.

Another study investigated the viral community composition in a watershed contaminated with microplastics (MPs) [50]. The study found diverse viral communities in MP samples collected upstream of the estuary [50]. An interesting aspect about this study is that MPs have not been largely explored as vectors of viruses in water sources using viral metagenomics; thus, the results contribute to current knowledge of potential vectors of viruses, particularly those originating from fecal material. Expectedly, the study found that the dominant viruses in the watershed were bacteriophages (including myoviruses, siphoviruses, and podoviruses), and >900 viral species were shared along the river [50]. Notably, these viral families are known to be common in both environmental and fecal samples. MPs found upstream and downstream of the estuary seemed to be the major sources of viruses, while sewage did not seem to be the major source of viruses in MPs. The authors also found human herpesviruses in polypropylene (PP), which is intriguing and merits further investigation. While the study did not find significant sources of enteric viruses, similar studies could be performed in watersheds impacted by various sources of fecal contamination. Not surprisingly, from the sequences found to be of potential viral origin, only 22.8% were assigned to known viruses [50]. These results open the opportunity to further explore the identity and role of the viral dark matter in water sources contaminated with MPs. 

Another study evaluated the presence and abundance of crAssphage and other viruses in metagenomic datasets originating from various sources, including marine water [42]. crAssphages, PMMoV, adenoviruses, polyomaviruses, Torque teno viruses, and noroviruses were among the viruses investigated in the marine metagenomes. Interestingly, these viruses were not identified in the marine metagenomes included in this study, possibly due to a lack of sensitivity, inadequate sequencing depth, and/or these marine samples may have not been impacted by fecal material. Nevertheless, more studies of this nature are needed to understand the potential use of viral metagenomics to assess fecal contamination originating from various sources in marine waters. These studies also highlight that viral metagenomics has enabled a broader look into marine and freshwater environments and revealed connections from a One Health perspective that would not have been possible otherwise. 

### 3.2. Wastewater and Fecal Material

Wastewater and fecal material originating from various sources are among the main contributors of enteric pathogens, including viruses, in water sources used for drinking, recreation, and irrigation, as well as foodstuff and other sample types. Thus, several studies have also investigated the potential of viral metagenomics as a tool to track enteric viruses in wastewater (Figure 2). A study evaluated the presence and abundance of crAssphage in metagenomic samples originating from sewage, reclaimed water, freshwater, hypersaline water, and marine water [42], and identified crAssphages, as well as PMMoV, adenoviruses, polyomaviruses, Torque teno viruses, and noroviruses in the sewage and fecal samples [42]. Interestingly, crAssphage were identified in sewage samples as well as bat guano [42], suggesting multiple potential hosts for crAssphage and crAss-like phage, and prompting a number of questions around the origin(s) and potential transmission of this bacteriophage. In addition, such studies suggest the opportunity to further explore publicly available metagenomic datasets to identify known viruses with the potential to be used to track sources of fecal contamination. Alternatively, data analyses, similar to those leveraged in the discovery of crAssphage, may open the opportunity to identify other novel, host-specific enteric viruses in wastewater and fecal material. 

Another study looking into wastewater samples in a large city in Latin America applied viral metagenomics, followed by metagenome assembly, prediction of protein-coding regions, and mapping against the NCBI non-redundant (nr) database to characterize the RNA virome [51]. Results showed the high prevalence of double stranded RNA (dsRNA) viruses with the potential to infect bacteria, invertebrates, and humans in the wastewater samples. Picobirnaviruses and rotaviruses were also highly prevalent in the sewage samples. While this study focused on the applications of viral metagenomics as a tool for public health surveillance, similar approaches may provide information on the specific sources of enteric viruses present in wastewater samples [51]. 

Wastewater has also been explored as a viral reservoir. A variety of virus types, including respiratory viruses, are known to be excreted through fecal material and possibly reach wastewater systems [52,53,54], and viral metagenomics has been used to determine the presence of various viruses from a fecal and respiratory origin in wastewater and sewage sludge. By using metagenome assembly, tBLASTx, and mapping in sewage sludge samples, most of the reads that could be attributed to known viruses were phage, human pathogens, and other eukaryotic viruses [55]. The results of this same study also showed that both DNA and RNA viruses could be identified, and several of them included adenoviruses, astroviruses, and coxsackieviruses. Interestingly, herpesviruses and papillomaviruses were among the most prevalent, likely due to the shedding of these viruses through urine [56,57]. Similarly, respiratory viruses, including coronaviruses and rhinoviruses were also identified in the sewage sludge samples [55], and during the recent SARS-CoV-2 pandemic, tremendous effort was put towards the detection of SARS-CoV-2 in wastewater in order to monitor community-level infection rates and predict future outbreaks. Multiple studies have demonstrated that SARS-CoV-2 loads in wastewater can serve as one-to-two-week leading indicators of community case load [58,59,60,61]. Similarly, viral surveillance of wastewater, as a tool to monitor community-level health trends, recently confirmed the detection of poliovirus in both the United Kingdom and New York state, as well as the Mpox (formerly known as monkeypox) virus in non-endemic locations with no history of previously reported cases [62,63,64]. While these were detected using PCR-based methods, viral metagenomics has been successfully applied in wastewater to detect viruses belonging to the *Poxviridae* family [65].

### 3.3. Foodstuff and Soil

Foodstuff may become contaminated with fecal material through direct contact, exposure to runoff, irrigation with fecally-contaminated waters, or propagation in contaminated soils [66]. Clinical symptoms of ingesting contaminated food items are similar to those associated with exposure to and ingestion of fecally-contaminated water including diarrhea, and vomiting. Surveillance of viruses spreading through contaminated food items may usually start with the reporting of the disease(s) in hospital settings [67], which usually occurs when patients present severe symptoms and self-help is no longer sufficient. Epidemiological investigations typically occur after a group of individuals have reported similar symptoms after ingesting foodstuff originating from the same source [67]. Unfortunately, it is not always straightforward to pinpoint the original source of a foodborne outbreak. Both bacteria and viruses can cause foodborne disease, with noroviruses being among the top causes, followed by *Campylobacter*, and hepatitis A and E viruses [67,68]. While most foodborne diseases and outbreaks originate from human fecal material, some may also be caused by zoonotic pathogens. Thus, it is important to consider local and regional agricultural and water management practices before ruling out foodstuff as a source of human infection and outbreak. For instance, it is known that Nipah viruses can be transmitted from bats to humans through contamination of date palm sap as it is collected in open containers that bats have access to [69]. A more recent example is the potential origin and transmission of SARS-CoV-2 from animals to humans through the handling and/or consumption of wildlife [70]. Although SARS-CoV-2 is a respiratory virus, it is known to be associated with diarrhea in certain patients, and excreted through feces [71]. Indeed, it is known that certain animal food sources may increase the risk of disease spillover from wild animals and/or animal farming by direct contact and/or consumption [72]. 

While tracking the source of fecal contamination in foodstuff has been historically performed using culture and/or molecular methods targeting specific viruses, recent studies have also explored the use of viral metagenomics for such purposes (Figure 2). Advances from viral genome sequencing have also enabled the tracking of potential sources of fecal contamination in foodstuff using viral metagenomics, in part because no a priori knowledge is needed, and ample information can be obtained from such approaches (e.g., genetically, and phylogenetically). Indeed, viral metagenomic sequencing is suitable for the screening of foodstuff where the potential origin and viral content are unknown [67]. For instance, a study assessing the viral composition of commercial romaine and iceberg lettuces in fields and a distribution center applied viral metagenomics targeting both RNA and DNA viruses, and demonstrated that commercially grown and processed lettuce harbors an immense assemblage of viruses that infect a wide range of hosts, with plant pathogenic viruses dominating the viral fraction. [73]. Contigs belonging to human and animal viral pathogens from the *Reoviridae* and *Picobirnaviridae* families were also detected in all samples. Papillomaviruses were also identified in one field-grown iceberg lettuce sample, which showed a nucleotide identity of 98% to human papillomavirus type 3 and 100% of amino acid identity to the partial late protein gene. Moreover, viruses infecting bacteria, invertebrate, amoeba, fungi, and algae were also identified in the lettuce samples. Bacteriophages identified were associated with over 60 different bacterial hosts, including those that may have *Pseudomonas*, *Escherichia*, *Salmonella*, and *Vibrio* as the bacterial hosts, many of which are known to be potential pathogens [73]. Products derived from other food source derivatives, such as fetal bovine serum and trypsin, are also known to contain a myriad of viruses from various families including, but not limited to, *Parvoviridae, Anelloviridae, Flaviviridae*, and *Herpesviridae*, all of which have been identified using viral metagenomics [74].

Although viral metagenomics is not typically applied to soils in the context of tracking sources of fecal contamination, viral metagenomics has been used to understand the soil virome composition, ecology, and responses to land use and land cover change, as summarized by [75,76,77]. Indeed, soil viruses are increasingly recognized for their tremendous diversity, roles in shaping ecological dynamics in soil, and ability to rapidly reflect the influence of biotic and abiotic factors. Healthy soils are essential for agriculture, to promote plant and animal productivity and health, and maintain or enhance water and air quality. With the decreasing quality and health of soils globally, it is essential to improve our understanding of soil ecology, including the roles played by viruses, and maintain soil health. For this reason, viral metagenomic studies of soil samples will be a valuable approach moving forward, whether fecal contamination is the issue, or a broader understanding of soil ecology is needed to allow stakeholders to take timely decisions to make land management decisions and avoid public health concerns and economic loss.

### 3.4. Air

While fecal contamination is typically associated with water, soil, and food, air may also be affected by this type of contamination mainly through bioaerosols. The major bioaerosol sources of microorganisms, including viruses, in indoor and outdoor environments are humans, animals, ventilation systems, dust, and water used for cleaning, sanitizing, or irrigation [78]. Indoor environments, for example, are known to carry approximately 10^5^ bacterial and fungal cells per cubic meter of indoor air [79]. However, significantly less is known about the microbial quality of indoor and outdoor air samples, compared to other sample types such as water sources, foodstuff, and soil, and how potential sources of contamination can be traced using viral metagenomics. Indeed, sampling air may be more challenging compared to other sample types, as collection methods and collection efficiencies can vary from study to study [80]. Historically, methods for air quality analysis have been limited to bacterial, fungal, and viral cultures, but more recently they have begun to include metagenomics in order to capture the unculturable microbiome [80]. Yet, while most studies have focused on the bacterial and fungal fractions of air samples, several studies have explored the use of viral metagenomics to identify both pathogenic and non-pathogenic viruses (Figure 2). One caveat of such studies is the expected low biomass of air samples, which can result in low nucleic acid yields, and in turn can influence viral particle sequencing, analysis, and data interpretation. 

To address low nucleic acid yields, studies often employ a variety of methods to increase potential viral signals. One such study evaluating the virome composition of dormitory rooms through heating, ventilation, and air conditioning (HVAC) filters applied a number of steps, including cutting the filters into small pieces, suspending them in buffer, and sonicating them to detach viral particles before ultracentrifugation, nuclease treatment, and nucleic acid extraction [81]. Notably, as with any low biomass sample, a negative control was included to ensure no environmental contamination [81]. Results showed that one third of the dormitory rooms were characterized by having phage from a variety of putative bacterial hosts that seem to originate mostly from skin (e.g., *Pseudomonas* and *Propionibacterium*), the outdoor environment (e.g., *Sinorhizobium*), and fecal material (e.g., crAssphage). Insect, plant, and animal viruses including, but not limited to, densoviruses, tymoviruses, and retroviruses were also identified [81]. While the identities of several of the viruses (as assessed via BLASTx) were low (e.g., the lowest being 47%), it should be noted that such approaches aim to capture as many viruses as possible, with the caveat of increasing the false positive rate. Importantly, this study demonstrates that viral metagenomics can be applied to indoor air samples to track specific sources of contamination, including fecal contamination.

Another study assessed the viral content of farm dust [82], as farm animals may harbor viral pathogens, which in turn can be carried in dust, transmitted to humans and other animals, and consequently cause disease. Specifically, the authors explored the viral content of chicken farm dust in parallel with that of chicken feces at various time points. In this study, farm dust samples were collected using electrostatic dustfall collectors. Results showed that picornaviruses were highly prevalent in both dust and chicken feces, followed by parvoviruses, astroviruses, and caliciviruses. Chicken farm dust samples, particularly, were characterized by having DNA virus contigs sharing homology to adenoviruses, and circoviruses [82]. This study highlights the application of viral metagenomics to track sources of fecal contamination in dust, and further supports the idea that viruses are transferred between animals and their environment through air. 

### 3.5. Clinical Settings

While wastewater originating from domestic, industrial, and agricultural sources represent a concern to public health, hospital wastewater has the potential to be a highly diverse source of human pathogenic viruses. Although sources of viruses in hospital effluent originate primarily from human fecal material, urine, and other bodily fluids, the presence of specific viruses in hospital effluent can be utilized as indicators for epidemiological purposes, and the identification of emerging and re-emerging viruses. Adenoviruses, astroviruses, noroviruses, hepatitis viruses, and rotaviruses are known to be present in hospital effluent, representing a potential risk to public health, and have been mainly identified using molecular-based methods [83,84]. In addition to molecular-based methods, recent studies have applied viral metagenomics to characterize hospital wastewater, and have found that the majority of the viruses captured with the applied sequencing depth and technique (i.e., no a priori concentration step for viruses) were bacteriophages [85]. This demonstrates that concentrating viruses from wastewater samples and determining the appropriate sequencing depth are important factors to consider when addressing the presence of pathogenic viruses in hospital wastewater, since these may be present in low concentrations. Another study identified several pathogenic viruses in wastewater, using both PCR-based techniques and viral metagenomics, many of which were also identified in patients [86]. Several of the viruses identified using both molecular and sequencing techniques included, but were not limited to, adenoviruses and noroviruses [86].

Similar to hospital wastewater, air from inpatient or outpatient wards could potentially lead to hospital-acquired infections. While most of the viruses in the air from patient wards are expected to be of a human origin, this sample type can be particularly helpful when identifying emerging and re-emerging viruses. Similar to surveying viruses in hospital wastewater, most methods utilized in air samples originating from patient wards rely on the amplification of specific genes and genomic regions [87]. Varicella-Zoster [87], corona [88], and Mpox viruses [89], have all been detected in patient wards using PCR-based methods. As with air samples originating from dorms and farms, discussed above, collecting air samples originating from patient wards represents a challenge mainly due to the low biomass nature of the samples, and the variability in collection methods [90]. Viral metagenomics has not been extensively employed as a tool to characterize viruses in patient wards, and thus remains to be further evaluated. 

As mentioned, viruses have long been considered indicators of various sources of fecal contamination. Table 1 highlights several non-pathogenic and pathogenic viruses often used to track sources of fecal contamination. Although the list does not cover all currently used and proposed viral indicators of fecal contamination, it highlights those identified using viral metagenomics. The table also emphasizes the application of the shown viruses (i.e., indicator vs. pathogen) (Table 1). 

## 4. Considerations and Limitations of Viral Metagenomics for Microbial Source Tracking

As with other metagenomic-based approaches for tracking sources of contamination, viral metagenomics is not without limitations and potential sources of bias, each of which may influence one’s downstream results and interpretation. Such factors include sample processing, viral nucleic acid recovery, sequencing depth, sequence specificity, and bioinformatic tools, each of which should be considered and best matched to one’s study design and goals. These are described in more detail in the following section and table (Table 2). 

### 4.1. Sample Processing

Sample processing is a major consideration and influencing factor in viral metagenomic studies, and the techniques involved in sample processing differ depending on whether one intends to sequence nucleic acids obtained from whole samples or isolated viruses and VLPs (Table 2). The sequencing of nucleic acids from whole samples requires less preprocessing and specialized equipment, but it typically results in the recovery of host and bacteria, which results in a small proportion of resultant sequence reads being attributed to viruses. The implication of this is that the viral fraction detected using shotgun metagenomics of the whole sample will correspond to the highly abundant and common viruses, some of which are expected to be integrated into the host’s (e.g., bacteria) chromosome. On the other hand, VLPs isolation, prior to sequencing, enables the identification of both high and lower abundance viruses since debris, host, and bacterial cells are removed through a series of filtration and concentration steps, but it is more labor intensive and, in some cases, cost-prohibitive. In order to isolate VLPs, samples typically are first centrifuged to remove large debris, followed by sequential filtration through 0.45-μm and 0.2-μm filters to trap cells, and the filtrate is then further processed, if desired. VLP purification steps may differ depending on the sample type, but can involve a cesium chloride (CsCl) density gradient ultracentrifugation step, and a DNase/RNase treatment prior to nucleic acid extraction to ensure that no host and bacterial nucleic acids influence results (this is especially important for the study of bacteriophages) [91]. While methods for VLPs isolation for viral metagenomics were originally developed to study marine viromes [32], the methods have been benchmarked for various sample types including seawater [32], soil [92], fecal material [93], and wastewater [94], and have been evaluated in multiple studies [95,96].

### 4.2. Nucleic Acid Recovery

After identifying the most suitable approach to sample processing for one’s project goals, the next consideration is the type(s) of nucleic acid to target. This may be dependent on sample type and may also influence the selection of the methods and protocols for nucleic acid extraction [97] (Table 2). For instance, marine waters, fecal material, and wastewater are often dominated by DNA viruses (primarily bacteriophages), while eukaryote-infecting viruses are frequently RNA viruses. The target nucleic acid will also influence the approach for the purification, concentration, and sequencing methods [97]. For instance, VLP isolation is suitable for both DNA and RNA viruses of varying abundance, but total DNA or RNA isolation typically limits one’s view to the most abundant viruses in the sample matrix, and may not necessarily capture low abundance and/or transient viruses [97]. In the context of tracking sources of fecal contamination, the isolation of viruses and VLPs may provide greater odds of the detection of enteric pathogenic viruses (versus whole sample nucleic acid extraction) given that they are typically present in low concentrations in environmental samples, such as freshwaters and marine waters. Indeed, it is known in environmental microbiology that concentration of the viral fraction is essential to reach a suitable level of sensitivity, and some methods may be more efficient than others for viral recovery [98]. 

Following the selection of the sample processing approach (whole sample vs. VLPs) and target nucleic acid(s) (DNA vs RNA, or both), the next step is to ensure an efficient viral nucleic acid extraction (Table 2). The total nucleic acid yield of any sample type will be dependent on the sample’s quantity and volume, initial virus concentration, and the effectiveness of the extraction method. A study evaluated nucleic acid extraction efficiency by testing eight different commercially available kits using liquid biosamples (e.g., blood, tissue, and cell-free body fluids) [99]. The kits in the study included those targeting DNA or RNA viruses, or both, mainly from blood, tissue, and cell-free body fluids. The protocols of several of these commercial kits are very specific to the reagents comprising the kit, but one of the main differences in the extraction of a specific virus (DNA vs RNA) is the type of nuclease used. For instance, to target DNA or RNA viruses, usually an RNAse or DNAse will be added after extraction, respectively. This particular study showed that any extraction kit that is performing satisfactorily for viral PCR diagnostics can be used for shotgun sequencing for the identification of viruses in liquid specimens [99]. Another study showed that a specific commercial kit provided the best results for DNA viruses in respiratory samples in terms of Ct values compared to the other extraction kits included; another kit provided the best extraction efficiency for shotgun metagenomic sequencing [100]. Since viruses usually need to be concentrated from environmental samples, particularly freshwater and marine waters, the nucleic acid extraction method will depend on the resulting concentrate. In some cases, the resulting concentrate can be in liquid or solid form, depending on the concentration method used. Similarly, specific nucleic acid recovery methods may also apply to foodstuff and air samples due to the nature of the samples and include, but are not limited to, direct extraction and/or proteinase K treatment [101]. Nucleic acid recovery in foodstuff and air samples depends, in turn, on how the samples are collected and how viruses are separated from the matrix and concentrated prior to nucleic acid recovery. Virus separation methods include elution techniques such as washing the viral particles from the matrix using appropriate buffers [101]. Virus concentration methods include, but are not limited to, Polyethylenglycol (PEG) precipitation, evaporation, ultracentrifugation, ultrafiltration, and flocculation, and may apply to different samples types such as water, foodstuff, and air samples [101]. 

### 4.3. Sequencing Depth

After sample processing and nucleic acid extraction, samples go through library preparation and sequencing. An important aspect during this process is determining the required number of sequences needed to obtain a virus representation in an unbiased manner (Table 2). While this has been determined in studies targeting the 16S rRNA gene of bacteria and archaea where few thousands of reads are needed to achieve a similar diversity compared to deeper sequencing, depending on the sample type [102], as well as studies looking at the group of ARGs, also known as the resistome [103], fewer studies have determined the number of reads needed in viral metagenomics to obtain an unbiased representation of the viral communities in a given sample. The sequencing depth needed will be dependent on the sample type, host content, and if a virus and VLPs concentration step(s) was applied. For instance, a study looking at the human urinary virome in association with urinary tract infections (UTIs), and where viruses and VLPs were concentrated using CsCl gradient ultracentrifugation, showed that 10,000–20,000 sequences were sufficient to achieve an unbiased representation of the human urine virome [57]. In fecal samples where shotgun metagenomics of the whole sample has been applied, it has been estimated that over 15 million reads are needed to achieve a representation of the expected viruses [104]. Expectedly, these studies show that sequencing depth is essential when determining virome composition in various sample types, and that it is an important component when considering viral metagenomics as a tool for tracking sources of fecal contamination. Since viral metagenomics does not target any specific virus, there is also the likelihood of detecting the autochthonous virome of a given sample, which may limit the detection of certain enteric pathogenic viruses, particularly those present in low abundances. For this reason, determining the optimal sequencing depth is essential in virome research, and when using viral metagenomics to track sources of fecal contamination. 

### 4.4. Bioinformatic Analysis

Tracking sources of fecal contamination using viral metagenomics requires sequences to share a high degree of identity with known viral reference genomes, typically those originating from sources of interest, such as humans or livestock. This, in turn, requires the use of bioinformatic tools that are highly specific and sensitive. Bioinformatic tools for virus sequence annotation that have been developed specifically for virome analysis are, generally, alignment-based, and include FastViromeExplorer [105], PHASTER [106], and VirMAP [107], or k-mer based and include MetaVir2 [108] and VirFinder [109]. Machine learning algorithms for virus classification such as DeepVirFinder [110] have also been developed (Table 2). The selection of the virus annotation tool and database for the analysis of viral metagenomics data can influence the interpretation of the results. For example, in the case of alignment-based analyses, which are generally database dependent, viruses that are not present in the database may go unannotated, or may be misassigned to another closely related virus [111]. Similarly, assembly-based methods may fail to detect viruses in cases where insufficient read depth, low read coverage, and genomic repeats lead to poor and/or fragmented assemblies and result in poor genome recovery [45]. One potential way to circumvent biases associated with virome assembly and annotation is by adding a viral mock community, composed of whole viruses, nucleic acids, or synthetic genomes, as well as other similar controls, into the analysis. The addition of a ground truth sample(s) is essential when determining the most suitable assembly and annotation tool for viral sequences in any sample type [35]. These types of controls aid in determining the specificity and sensitivity of viral assembly and annotation tools by looking at true positive/false negative rates. This approach may be valuable when using viral metagenomics for tracking sources of fecal contamination.

An intrinsic challenge associated with viral metagenomics used for any purpose, including tracking sources of fecal contamination, is the lack of similarity of the sequences to known viruses (Table 2). Indeed, up to 99% of the sequences expected to be of viral origin cannot be taxonomically classified in certain sample types due to the high degree of sequence divergence [112]. This viral fraction lacking similarity to known viruses, and for which additional bioinformatic and cultivation efforts are needed, is known as the viral dark matter [33,113], and studies seeking to provide greater insight into the viral dark matter are published on a routine basis. Such studies have resulted in the discovery of crAssphage [36], and other viruses and viral groups, including corona-like viruses [114], redondoviruses [115], and candidate viral families “Quimbyviridae”, “Flandersviridae”, and “Gratiaviridae” [116]. crAssphage represents one of the most evaluated indicators of human fecal contamination to have emerged from studies of the viral dark matter. Due to the presence of crAssphage in wastewater, it has been proposed as a potential indicator of human fecal contamination. Interestingly, recent studies have challenged this by identifying crAssphage and its bacterial host in feline fecal samples [117]. This illustrates the need to further authenticate a potential viral indicator of specific sources of fecal contamination by evaluating its presence and prevalence in various fecal sources originating from different animal hosts. While developing novel viral indicators as specific sources of fecal contamination is challenging given the need for robust, sensitive, and specific primers for rapid screening [118], bioinformatic and culture pipelines, such as those applied in the discovery of crAssphage, should precede the validation of a viral candidate as an indicator of specific sources of fecal contamination due to their potential to produce high-resolution broad-level snapshots of the microbial and viral profiles within the environments of interest.

## 5. Future Directions of Viral Metagenomics in One Health

The present review highlights the need for more studies applying viral metagenomics to track specific sources of fecal contamination in various environments and sample types such as marine waters, freshwaters, foodstuff, and air. In addition, it is evident that there is the need for more studies assessing the mentioned considerations and limitations often encountered in viral metagenomics, including, but not limited to, sample processing, nucleic acid extraction, sequencing depth, and bioinformatic analyses when tracking sources of fecal contamination. These considerations and limitations also influence the development of protocols aiming to characterize potential novel viral indicators of fecal contamination, as well as the associations of viruses identified through viral metagenomics with risk factors such as the presence of specific pathogens and ARGs. Current and future practices and events leading to fecal contamination (e.g., sewage overflows, septic tank, and pipes leakages) may impact water sources, foodstuff, soil, and air, which ultimately can affect One Health—human, animal, and environmental health, and the linkages among them. Viral metagenomics is already being applied to track sources of fecal contamination in waters to identify specific enteric viral pathogens of concern, and to develop PCR-based assays targeting specific viruses, as in the case of crAssphage [119]. Indeed, viral metagenomics will continue to enable tracking specific sources of fecal contamination in other sample types, and could potentially be used for such purposes in sample types that have not been fully evaluated for these purposes. Besides tracking sources of fecal contamination, viral metagenomics continues to be explored in other areas important in One Health including surveillance of emerging and re-emerging pathogens, the potential application of machine learning algorithms, and improvement of quantitative methods and relative quantitative methods in viral metagenomics. In fact, several studies have evaluated this approach during the SARS-CoV-2 pandemic to identify current and potentially new variants in wastewater samples [15,120]; thus, viral metagenomics possesses the potential to be part of the toolbox of methods enabling a One Health approach.

## Figures and Tables

**Figure 1 viruses-15-00236-f001:**
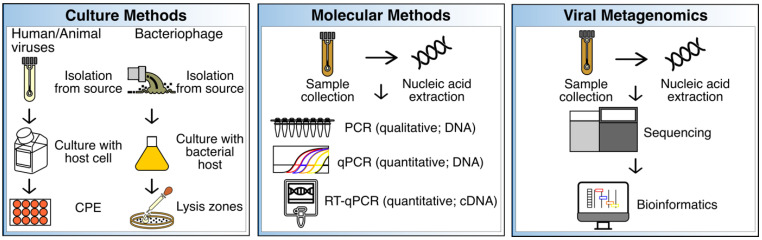
Summary of culture, molecular, and viral metagenomic methods. These methods have been benchmarked to track sources of fecal contamination in various sample types (see text). CPE = cytopathogenic effects; qPCR = quantitative PCR; RT-qPCR = reverse transcription qPCR; cDNA = complementary DNA.

**Figure 2 viruses-15-00236-f002:**
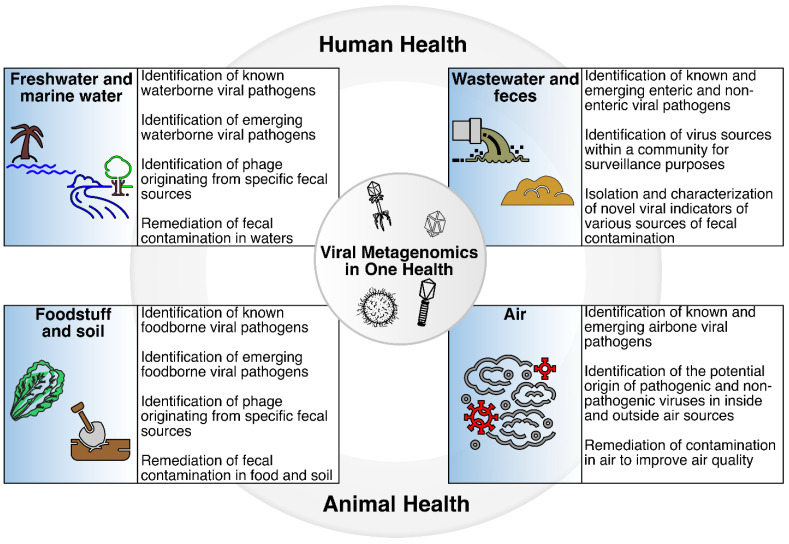
Viral metagenomics in One Health. The introduction of fecal material into water sources, foodstuff, soil, and air has the potential to affect One Health, particularly human and animal health, through exposure to enteric pathogens including viruses. Viral metagenomics may be part of the suite of methods that can be used to identify known and emerging viruses, track sources of fecal contamination, identify new viral indicators of specific sources of fecal contamination, and remediate sources of contamination in various sample types.

**Table 1 viruses-15-00236-t001:** Examples of viral family/species/types detected in water, soil, foodstuff, and air using viral metagenomics. Table highlights those viruses with indicator potential or potential pathogenic/health threat.

Virus	Virus Potential	Reference
Coliphages	Indicator	[22]
Pepper Mild Mottle virus	Indicator	[42]
crAssphage	Indicator	[42,43]
*Bacteroides* phage (other than crAssphage)	Indicator	[44]
Torque teno virus	Indicator	[42,55]
Polyomaviruses	Pathogen	[42]
Picobirnaviruses	Pathogen	[51]
Astroviruses	Pathogen	[55]
Coronaviruses	Pathogen	[55]
Coxsackieviruses	Pathogen	[55]
Hepatitis viruses	Pathogen	[55]
Herpesviruses	Pathogen	[55]
Adenoviruses	Pathogen	[55]
Papillomaviruses	Pathogen	[55]
Rhinoviruses	Pathogen	[55]
Rotaviruses	Pathogen	[51,55]
Poxviruses	Pathogen	[65]
Reoviruses	Pathogen	[73]
Caliciviruses	Pathogen	[82]
Picornaviruses	Pathogen	[82]
Parvoviruses	Pathogen	[55,74,82]
Noroviruses	Pathogen	[86]

**Table 2 viruses-15-00236-t002:** Considerations and limitations of specific steps of a viral metagenomic pipeline. These considerations and limitations may also apply to the use of viral metagenomics when tracking sources of fecal contamination.

Step	Considerations	Limitations
Sample processing	Sequencing whole sample	Viruses will represent a small fraction of the sequences Missed opportunity to study low abundant and transient viruses Additional steps for virus isolation may be cost-prohibiting and time-consuming
	Isolating viruses and virus-like particles (VLPs)	Specific isolation methods and protocols may be biased towards certain viral groups, and may miss the opportunity to study viruses that are integrated
Nucleic acid recovery	Targeted nucleic acid (i.e., DNA, RNA, or both)	May be dependent on the sample type (e.g., fecal material and water), and targeted host (e.g., bacteria and eukaryotic cells)
	Extraction method/protocol/kit	May be optimized for specific sample types and a specific nucleic acid
Sequencing	Sequencing depth	Certain viruses may not be detected if an optimal sequencing depth is not reached
Bioinformatic analysis	Alignment- vs. k-mer- vs. Machine-learning-based tool	True positives, false positives, and false negatives may be dependent on the bioinformatic tool used A large fraction of the sequences may not match known viruses when the bioinformatic tool is database-dependent
	Analysis of reads or assembled reads	Computational resources required may vary if reads or assembled reads are used for downstream analysis
	Addition of a ground truth sample (e.g., mock community)	Ground truth samples may not always reflect the composition of a true sample

## Data Availability

Not applicable.

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
