# Peer review of "Viral Metagenomics as a Tool to Track Sources of Fecal Contamination: A One Health Approach"

_viruses, 2023, doi:10.3390/v15010236_

Round 1

Reviewer 1 Report

I think it is a good review, there are many examples and most of the techniques used in metagenomics are described, but I would like to know why you did not mention the Single Virus Genomics technique. It is my only complaint

Author Response

I think it is a good review, there are many examples and most of the techniques used in metagenomics are described, but I would like to know why you did not mention the Single Virus Genomics technique. It is my only complaint.

We thank the reviewer for this important input. We agree that the Single Virus Genomics technique is very important in the field and will be used in the field of microbial source tracking using viruses. We have now added more information about this technique in the revised version of the manuscript (lines 193-217).

Reviewer 2 Report

In this review, the authors talked about the potential use of viral metagenomics to track fecal contamination, including its advantages and limitations. Viral metagenomics is being used in a variety of basic and clinical settings, including but not limited to viral discovery, ecology and diversity, clinical diagnostic, virus surveillance. This review focuses on its use in fecal contamination, which in main threat to the One Health.

Generally, this manuscript is well organized and deserved to be published. I only have a few minor suggestions that could be considered to further improve this review paper.

The author discussed fecal viral contaminations from water, soil, foodstuff and air, I would suggest they can search and discuss more on fecal viral contaminations from clinical settings, such as sewage water from hospital and air from inpatient/outpatient ward. As the fecal virus shedding in hospital sewage water can be indicators or clues for local epidemiology and emerging of novel viruses; viruses in air/aerosols from inpatient/outpatient ward could lead to hospital acquired infections.

Not quite sure if it is within the author’s initial focus of this review, but meat from many animals such chicken, pig and cattle could carry viral contaminations, possibly through fecal contamination during production or other routes. Other biological products such as serum and trypsin that are normally produced by farm animals are also reported to carry many viral sequences by viral metagenomics. 

As many viruses are presented in different sample types, it would be appreciated if the author could summarize the current knowledge of the viral family/species/types (could provide them in a Table) that are detected in water, soil, foodstuff and air. Highlight those viruses with indicator potential or potential pathogenic/health threat to animals and humans.

Author Response

The author discussed fecal viral contaminations from water, soil, foodstuff, and air, I would suggest they can search and discuss more on fecal viral contaminations from clinical settings, such as sewage water from hospital and air from inpatient/outpatient ward. As the fecal virus shedding in hospital sewage water can be indicators or clues for local epidemiology and emerging of novel viruses; viruses in air/aerosols from inpatient/outpatient ward could lead to hospital acquired infections.

We thank the reviewer for highlighting this. We have now expanded the review to discuss viral contamination in hospital sewage and air from patient wards (lines 474-508).

Not quite sure if it is within the author’s initial focus of this review, but meat from many animals such chicken, pig and cattle could carry viral contaminations, possibly through fecal contamination during production or other routes. Other biological products such as serum and trypsin that are normally produced by farm animals are also reported to carry many viral sequences by viral metagenomics. 

We thank the reviewer for this suggestion. While this was not within the initial focus of the review, we have added this information in the manuscript text (lines 386-388; 409-413).

As many viruses are presented in different sample types, it would be appreciated if the author could summarize the current knowledge of the viral family/species/types (could provide them in a Table) that are detected in water, soil, foodstuff, and air. Highlight those viruses with indicator potential or potential pathogenic/health threat to animals and humans.

We thank the reviewer for this suggestion. We have now included new Table 1 with examples of viruses that are used to track sources of fecal contamination or are pathogens.

Reviewer 3 Report

The manuscript brings a compilation of information regarding viral metagenomics as a tool to track sources of fecal contamination in a unique health approach. In general, the text of the manuscript is well written, the subjects were approached in a logical and succinct sequence. In addition, the figures and tables adequately summarize the information mentioned throughout the text of the manuscript. However, suggestions for improving the text of the manuscript were listed below in the reviewer's comment. Therefore, the manuscript can be accepted for publication after the suggested adjustments.

Author Response

Line 9: I suggest removing "various methods" and if possible cite them, if the list is long, cite the main ones.

We thank the reviewer for this suggestion. We have modified the abstract accordingly (line 9).

Line 15: ..."various sample types", as written seems a bit vague to me, rewrite the sentence.

We have removed “various sample types” for conciseness as the sample types are mentioned in the following sentence.

Line 22: I suggest using other keywords that have not been mentioned in the title of the manuscript.

We thank the reviewer for this suggestion. We have now added additional keywords not mentioned in the title.

Line 65: ..."various conditions" would be experimental conditions in the laboratory or experiments carried out in the field. It would be interesting to leave under which conditions the studies with fresh water were conducted.

We thank the reviewer for these suggestions. We have now modified the sentence to include the specific experimental conditions (lines 68-69).

Line 72: ..."type of matrix"... Here, are you referring to organic or inorganic compounds? This must be made clear.

We have now clarified that it is both organic and inorganic compounds and have added an appropriate reference (line 77).

Line 112: ..."various sources", be clearer.

We have clarified “various sources” as those coming from, but not limited to environmental, animal, and clinical samples (lines 117-118).

Line 130: ..."various", I suggest replacing it with another more appropriate term.

We have removed “various” from this sentence to avoid confusion and for conciseness.

Lines 141 to 142: ..."various"..., same comment made earlier.

We have removed “various” from this sentence to avoid confusion and conciseness.

Line 144: Specify what the molecular methods are or remove "certain" and adjust the sentence accordingly.

We have modified “certain” and modified the sentence accordingly.

Line 150: Remove "viral metagenomics".

We have removed “viral metagenomics”.

Line 178: ..."various"...remove and adjust the text accordingly.

We have removed “various” and modified text accordingly.

Line 180: ..."interrogated"... remove and adjust the text accordingly.

We have removed “interrogated” and modified the sentence accordingly.

Line 181: ..."various"..., same comment made earlier.

We have removed “various” and modified text accordingly.

Line 183: ......"various"...remove and adjust the text accordingly.

We have removed “various” and modified text accordingly.

Lines 182 to 186: What is the importance of this particular bacteriophage? What types of samples was this bacteriophage found in different species? What is the environment in which this bacteriophage was found and associated with humans? Please be clearer. If this bacteriophage has no assigned importance, I suggest mentioning the presence of bacteriophage in X, X and X samples from humans, animals, and the environment, respectively, for example. Remove "by using BLAST as the alignment tool" and adjust the text accordingly.

We thank the reviewer for pointing this. This bacteriophage mentioned in the review has been shown to infect B. fragilis strains that seem to be associated with humans; thus, have the potential to be used in microbial source tracking. We have clarified this in the revised version of the manuscript (line 189-190).

We have also removed "by using BLAST as the alignment tool" and adjusted the text accordingly.

Line 218: Define the "etc" or remove.

We have removed etc (line 269).

Lines 484 to 486: What are these other methods? It would be interesting to mention, it would provide more context to the manuscript.

We thank the reviewer for highlighting this. We have now expanded this section to add some of the other methods (lines 594-603).

Lines 521-522: How can the viral annotation tool influence the results? Be clearer.

We thank the reviewer for pointing this. We have now clarified this sentence by stating that one of the main reasons viral annotation tools can influence results is due to the nature of several of the annotation tools. For instance, alignment-based tools are database dependent, and many viruses not matching viruses in the database may go unannotated or misassigned to a closely related virus (lines 646-650). We have also clarified how assembly tools can potentially influence results (lines 652-653).

Line 530: Remove "positive/false".

We have removed accordingly (line 660).